# Clinical Value of the PD-1/PD-L1/PD-L2 Pathway in Patients Suffering from Endometriosis

**DOI:** 10.3390/ijms231911607

**Published:** 2022-10-01

**Authors:** Dorota Suszczyk, Wiktoria Skiba, Witold Zardzewiały, Anna Pawłowska, Karolina Włodarczyk, Grzegorz Polak, Rafał Tarkowski, Iwona Wertel

**Affiliations:** 1Independent Laboratory of Cancer Diagnostics and Immunology, Medical University of Lublin, Chodźki 4a, 20-093 Lublin, Poland; 2Students’ Scientific Association, Independent Laboratory of Cancer Diagnostics and Immunology, Medical University of Lublin, Chodźki 4a, 20-093 Lublin, Poland; 3I Chair and Department of Gynaecologic Oncology and Gynaecology, Medical University of Lublin, Staszica 16, 20-081 Lublin, Poland

**Keywords:** endometriosis, dendritic cells, programmed cell death pathway, peritoneal fluid, immunosuppression, peritoneal cavity

## Abstract

The interaction between dendritic cells (DCs) and T cells mediated by the programmed cell death 1 (PD-1)/programmed cell death ligand 1 (PD-L1)/programmed cell death ligand 2 (PD-L2) pathway is the most important point in regulating immunological tolerance and autoimmunity. Disturbances in the quantity, maturity, and activity of DCs may be involved in the implantation and growth of endometrial tissue outside the uterus in endometriosis (EMS). However, little is known about the role of the immune checkpoint pathways in EMS. In our study, we examined the expression of PD-L1/PD-L2 on myeloid DCs (mDCs) and plasmacytoid DCs (pDCs) in the peripheral blood (PB) and peritoneal fluid (PF) of both EMS patients (*n* = 72) and healthy subjects (*n* = 20) via flow cytometry. The concentration of soluble PD-L1 and PD-L2 in the plasma and PF of EMS patients and the control group were determined using ELISA. We demonstrated an elevated percentage of mDCs, mDCs and pDCs with the PD-L1or PD-L2 expression, and a higher concentration of the soluble forms of PD-L1 and PD-L2 in the PF than in the plasma of EMS patients. We conclude that the peritoneal cavity environment and the PD-1/PD-L1/PD-L2 axis may play an important role in the modulation of immune response and the development and/or progression of EMS.

## 1. Introduction

Endometriosis (EMS) is a chronic inflammatory condition defined by the presence and growth of ectopic endometrial-like tissue outside the uterus. It affects approximately 10% of women around the world [1]. The main symptoms of the disease include pelvic pain, dysmenorrhea, dyspareunia, gastrointestinal complications, and infertility. Recent reports suggest that EMS has the potential to transform into endometriosis-associated ovarian cancer (EAOC) [2,3]. It has been demonstrated that endometrial implants have similar characteristics to EAOC in that they involve an inflammatory condition, uncontrolled growth, the ability to invade surrounding tissues, as well as neoangiogenesis and a high risk of recurrence [4,5]. Despite a high prevalence of EMS, there are still some uncertainties about its pathogenesis, diagnosis, and therapy, emphasizing the need for further research into the essential pathophysiology.

It has been shown that what is important for the survival of endometriotic lesions is not only their resistance to apoptosis, but also their escape from immune surveillance. It is important to point out that endometriotic lesions are also characterized by a high proliferative rate that is stimulated via the inflammatory microenvironment [6]. Recent studies have shown that EMS is associated with changes in systemic and local immunity [7]. The underlying mechanisms include quantitative and functional disorders of neutrophils, monocytes/macrophages (MO/MA), natural killer (NK) cells, and T cells. A few reports have revealed that immunosuppressive cells, such as regulatory T cells (Tregs) and myeloid-derived suppressor cells (MDSCs), may promote the progression of EMS [8,9,10,11]. The unique microenvironment with the presence of MDSCs, tumor-associated macrophages (TAMs), and Tregs, which are the source of IL-10, have an impact on dendritic cells (DCs) maturation and the secretion of IL-12, and decrease their antigen presentation ability. IL-10 stimulates the conversion of immunogenic DCs into tolerogenic DCs and it may lead to the anergy of cytotoxic T cells [12]. Lymphocyte and NK-mediated cytotoxicity are one of the key mechanisms responsible for removing implants of endometrium in healthy women, whereas defective T-lymphocyte response is observed in patients with EMS [13].

The reduced cytotoxicity of NK and T cells can be an effect of immune checkpoints (ICs) expression on immune system cells. The programmed cell death ligand 1 (PD-L1) or programmed cell death ligand 2 (PD-L2) expression on antigen-presenting cells (APCs) may lead to inducing T cell anergy or apoptosis via programmed cell death 1 (PD-1) on T cells [14]. The co-inhibitory PD-1/PD-L1/PD-L2 pathway controls the central and peripheral T-cell tolerance by limiting the first phase of expansion or activation and restriction effector function of the self-reactive T cells. The pathway described above is also involved in the pathogenesis of autoimmune diseases, such as type I diabetes, multiple sclerosis, inflammatory bowel diseases (Crohn’s disease and ulcerative colitis), or rheumatoid arthritis [15], and in the pathogenesis of many cancers including OC [16,17,18,19]. The co-inhibitory PD-1/PD-L1/PD-L2 interaction contributes to inducing and maintaining iTregs, which suppress anti-tumor T cells response and contributes to tumor progression [15]. Moreover, a high level of PD-L1 expression correlates with the poor prognosis of OC patients [20,21,22]. 

Based on the research outcomes about the role of ICs in the tumor microenvironment or autoimmunological diseases, we hypothesize that the PD-1/PD-L1/PD-L2 pathway may play a significant role in the implantation and growth of endometrial tissue in the peritoneal cavity, and may be related to the local immunosuppression in patients suffering from EMS. 

The aim of our study was to investigate the prevalence of PD-L1, PD-L2 expression on antigen-presenting cells (e.g., myeloid and plasmacytoid DCs) in the peripheral blood (PB) and peritoneal fluid (PF) of women with EMS, and its correlation with clinical characteristics of EMS patients. We also evaluated the concentration of the soluble form of PD-L1 and PD-L2 in the plasma and PF, in terms of its clinical values in EMS progression. 

## 2. Results

### 2.1. Distribution of Myeloid and Plasmacytoid Dendritic Cells in Patients with Endometriosis

The investigation of dendritic cells in the systemic and local peritoneal environment of patients with endometriosis showed that the percentage of myeloid DCs was significantly higher (*p* < 0.0001) in the peritoneal fluid than in the peripheral blood. In contrast, the percentage of plasmacytoid DCs was significantly higher (*p* < 0.05) in the peripheral blood of EMS patients in comparison to the peritoneal fluid (Figure 1).

### 2.2. Percentage of Myeloid and Plasmacytoid Dendritic Cells in Relation to ASRM Stages of Endometriosis

Secondly, we examined the percentage of mDCs and pDCs subpopulations in PB and PF in early (I/II), and late (II/IV) *ASRM* stages of endometriosis. We did not observe significant differences (*p* > 0.05) in the percentages of mDCs and pDCs between early/late stages of EMS in both peripheral blood and peritoneal fluid (Figure 2).

### 2.3. Percentage of Myeloid and Plasmacytoid Dendritic Cells with PD-L1 or PD-L2 Expression in Patients with Endometriosis

In the next step, we examined the expression of PD-L1 or PD-L2 molecules on the surface of mDCs and pDCs in PB and PF of EMS patients. We observed that the percentage of both, mDCs and pDCs with PD-L1 expression was significantly higher (*p* < 0.001) in the peritoneal fluid than in peripheral blood of EMS patients. Similarly, we showed that the percentage of PD-L2 positive mDCs and pDCs was significantly higher (*p* < 0.001) in the peritoneal fluid than in the peripheral blood of EMS patients (Figure 3).

### 2.4. Distribution of Myeloid and Plasmacytoid Dendritic Cells in Patients with Endometriosis and the Control Group

We observed interesting results after comparing the percentage of mDCs and pDCs subpopulations in patients with endometriosis and healthy women. Our study showed that the percentage of mDCs was significantly higher (*p* < 0.05) in PB of patients suffering from endometriosis in comparison to healthy subjects. In contrary, the percentage of pDCs was lower in EMS patients than in PB of the control group; however, the difference was not statistically significant (Figure 4).

### 2.5. Percentage of Myeloid and Plasmacytoid Dendritic Cells with PD-L1 or PD-L2 Expression in Patients with Endometriosis and the Control Group

Next, we compared the percentage of mDCs and pDCs with PD-L1 or PD-L2 expression in the peripheral blood of patients with endometriosis with the percentage of these cells in PB of healthy women. We showed that the percentages of PD-L1 positive mDCs and pDCs in peripheral blood was significantly lower (*p* < 0.0001) in the EMS group in comparison to the control group. Similarly, the percentage of PD-L2 positive mDCs was significantly lower (*p* < 0.05) in the peripheral blood of EMS patients in comparison to the control group. The percentage of PD-L2 positive pDCs was lower in PB of EMS patients in comparison to the control group; however, the difference did not reach a significant level (Figure 5).

### 2.6. Percentage of PD-L1 and PD-L2 Positive Dendritic Cells in Relation to ASRM Stages of Endometriosis

Next, we determined the relationship between percentages of DCs with PD-L1 and PD-L2 expression and early (I/II) or late (II/IV) *ASRM* stages of endometriosis. There were no significant differences (*p* > 0.05) in the percentages of PD-L1/PD-L2-positive mDCs and pDCs in the peritoneal fluid between early/late stages of EMS. There were also no significant differences (*p* > 0.05) in the percentages of PD-L1-positive mDCs and pDCs, and PD-L2-positive mDCs in the PB between early/late stages of endometriosis (Figure 6).

Interestingly, we showed that the percentage of PD-L2 positive plasmacytoid DCs in peripheral blood was significantly higher (*p* < 0.05) in the late stages (III/IV) of endometriosis in comparison to early (I/II) stages of the diseases (Figure 6).

The percentage summary of PD-L1 and PD-L2 expression on mDCs and pDCs of EMS patients and the control group can be found in the table below (Table 1.)

### 2.7. Concentration of the Soluble Form of PD-L1 and PD-L2 in Plasma and Peritoneal Fluid of Patients with Endometriosis

The concentration of sPD-L1 in the peritoneal fluid (*n* = 36) was significantly higher (*p* < 0.05) than in the plasma (*n* = 61) of EMS patients. Similarly, the concentration of sPD-L2 was significantly higher (*p* < 0.001) in the PF than in the PB of patients with endometriosis (Figure 7).

### 2.8. Concentration of sPD-L1 and sPD-L2 in the Plasma and Peritoneal Fluid in Relation to ASRM Stages of Endometriosis

Interestingly, we demonstrated that the plasma sPD-L1 concentration in patients with late stages (III/IV) of endometriosis was higher than in early (I/II) stages of the disease and the difference reached the level of significance (*p* = 0.05). In contrast, concentrations of plasma sPD-L2 and peritoneal fluid sPD-L1 and sPD-L2 did not differ significantly between early (I/II) and late (III/IV) stages of endometriosis (Figure 8).

### 2.9. Concentration of sPD-L1 and sPD-L2 in Plasma of Patients with Endometriosis and the Control Group

Next, we compared the concentrations of sPD-L1 and sPD-L2 in the plasma of EMS patients and the control group. The concentration of sPD-L1 was significantly lower (*p* < 0.001) in the plasma of EMS patients (*n* = 61) than detected in the control group (*n* = 19). The concentration of sPD-L2 was also elevated in the plasma of EMS patients than in the control group; however, the difference was not statistically significant (Figure 9). 

### 2.10. Concentration of sPD-L1 and sPD-L2 in Plasma of Patients with Early (I/II) and Late (III/IV) ASRM Stages of EMS and the Control Group

In the end of analysis, we compared the concentrations of sPD-L1 and sPD-L2 in the plasma in the early (I/II) and late (III/IV) *ASRM* stages of EMS patients and in the control group. The concentration of both plasma sPD-L1 in the early (I/II) and late (III/IV) *ASRM* stages of EMS (*n* = 61) was significantly lower (*p* < 0.001 and *p* < 0.05, respectively) than detected in the control group (*n* = 19) (Figure 10).

On the other hand, the level of plasma sPD-L2 was significantly higher (*p* < 0.01) in the early (I/II) EMS stages than in healthy women (Figure 10). Additionally, the concentration of plasma sPD-L2 in the late (III/IV) *ASRM* stage of EMS patients was higher than in the control group; however, the difference was not statistically significant (*p* > 0.05).

## 3. Discussion

Although chronic inflammation and high estrogen concentrations are well-established characteristics of EMS, the etiology of this disorder remains unclear. One of the theories that could explain the disease development is an alteration in the immune system in terms of immune-cell recruitment, cell adhesion, and an upregulation of inflammatory processes, which can facilitate the implantation and survival of endometriotic lesions [23,24]. Despite advances in research on the immunological aspects of EMS, little is still known about the role of the immune checkpoint (PD-1/PD-L1/PD-L2) pathways in EMS. According to the current research, the expression of PD-L1 on the surface of DCs can inhibit the appropriate T cell activation and induce immunosuppressive Tregs development [25]. Due to systemic immunity, the interaction between DCs and T cells, mediated by the PD-1/PD-L1 pathway, seem to be the most important goal in regulating immunological tolerance and autoimmunity [25,26].

Against this background and given the enigmatic nature of EMS, we examined the expression of PD-L1 and PD-L2 on mDCs and pDCs in both the systemic and local peritoneal environment (i.e., PB and PF) of EMS patients. Moreover, we determined the concentration of the soluble form of PD-L1 and PD-L2 in the plasma and PF of EMS patients. The obtained data referred to the clinical characteristics of patients suffering from EMS and were compared with those obtained from healthy women. 

To the best of our knowledge, this is the first study on the expression of PD-L1 and PD-L2 on mDCs and pDCs in two different environments referred to as the clinical characteristics of EMS patients. It is worth highlighting that there is currently no evidence in the literature evaluating the soluble form of PD-L2 in patients with EMS.

Our study demonstrated some significant differences in DCs distribution in the two studied environments. The subsets of pDCs (BDCA-2^+^CD123^+^) were dominant in the PB, whereas mDCs (BDCA-1^+^CD19^−^) accumulated in the PF of EMS patients. The investigation of the PB samples obtained from healthy donors showed that the percentage of mDCs was higher in the EMS group, while pDCs were lower than those determined in the control group. On the contrary, Vallvé-Juanico et al., using mass cytometry, showed a higher proportion of pDCs in the peripheral blood of women suffering from EMS than in the control group. These contrary results might be a due assessment of pDCs in the proliferative phase of the menstrual cycle by the cited authors, and also different monoclonal antibodies used for phenotyping, or different cohorts of the patients. It should be stressed that authors evaluated pDCs in a very small study group (*n* = 13) of EMS patients versus the control group (*n* = 6). In our study, we evaluated DCs in numerous groups of patients with EMS (*n* = 62) and healthy women (*n* = 20) [27].

Similarly, Guo et al., using mass cytometry analysis, demonstrated a higher proportion of DCs (CD14^−^/CD11c^+^/HLA-DR^+^) in PF than in PB of EMS patients [28]. Additionally, Izumi et al. showed the accumulation of DCs in the PF of patients suffering from EMS [29]. It is worth noting that mDCs are a heterogeneous population of APCs responsible for the capturing and presentation of antigens to naïve CD4^+^ or CD8^+^ T cells leading to their activation. However, they have a dualistic role in their proinflammatory or regulatory properties modulated by microenvironmental conditions [26]. The tolerogenic role of mDCs is to control inflammation through hampering proinflammatory macrophages and T cells, and through the induction of tolerogenic Treg cells. An elevated percentage of Tregs was detected in the PF of EMS patients, and these are considered important mediators of tolerance during EMS development. Tregs also hamper the peritoneal immune response promoting immunosuppression [30,31,32]. They can inhibit the activity of T cells, decrease the phagocytic activity of macrophages, reduce cytotoxicity of NK cells, or cause disturbances in DCs’ maturation. Tregs may further decline the expression of CD80 or CD86 molecules on the surface of mDCs, diminishing the ability to present antigens and also inhibiting proper DCs development.

Fainaru et al. demonstrated that immature DCs stimulated the angiogenesis and growth of implants in a murine model of EMS. They showed that most of the CD11c+ DCs which infiltrated implants expressed vascular endothelial growth factor receptor 2 (VEGFR2), which is the receptor for pro-angiogenic factors, e.g., vascular endothelial growth factor (VEGF) [33]. Pencovich et al. showed that DCs stimulated the angiogenesis and growth of EMS lesions, and the ablation of DCs led to a significant decrease in their size in murine models of EMS. Another research group observed that pDCs are the source of IL-10 which stimulate angiogenesis in the murine model during the early stages of EMS [34]. Schulke et al. showed that the frequency of the mature CD83^+^-positive DCs was significantly decreased in eutopic and ectopic endometrium in women with EMS [35]. 

The elevated infiltration of pDCs was observed in several types of cancer, including melanoma, prostate cancer, head and neck cancer, breast cancer, and OC [20,36,37]. It has been documented that the circulating pDCs were increased in patients with gastric cancer and positively correlated with lymph node metastasis or advanced stages [36]. In our previous research, we demonstrated significantly higher percentages of mDCs and pDCs in PF than in PB, and in the tumor tissue of OC patients [20]. It should be stressed that EMS is often compared to cancer, and it is also known as a precursor to several types of OC [7,38,39,40].

It is worth mentioning that the presence of pDCs has also been documented in autoimmune diseases such as alopecia areata, systemic lupus erythematosus, systemic sclerosis, and psoriasis. The latest evidence suggests that DCs may be promotors of self-tolerance and may also be implicated in the development of autoimmune diseases [30]. In addition, pDCs are considered to be the source of interferon alpha (IFN-α), which promote the peripheral tolerance breakdown by the activation of immature mDCs or autoreactive B or T cells, and may lead to autoantibody production during autoimmune diseases [41].

To summarize, disturbances in the quantity, maturity, and activity of DC subsets may be involved in the implantation and growth of endometrial tissue outside the uterus by non-efficient clearing of endometrial cells, the induction of tolerance, and/or immunosupression.

In the present study, we also evaluated the expression of PD-L1 and PD-L2 on mDCs and pDCs, and the soluble forms of these ligands in the two environments of patients with EMS and healthy women. Interestingly, our study demonstrated the accumulation of mDCs and pDCs with the PD-L1 and PD-L2 expression in the PF of EMS patients. Moreover, we observed that the percentage of PD-L1 positive mDCs and pDCs in PB was significantly lower in EMS patients in comparison to the control group. It is worth highlighting that the decline of PD-L1 by chronic inflammation may lead to uncontrolled T cell proliferation which results in elevated autoimmunological T cell response [26]. According to some studies, EMS has some features an autoimmune disease manifested by tissue damage and the production of autoantibodies (against endometrium, histones, ovary, and phospholipids), and may be associated with other autoimmune diseases [23,42,43].

Wu et al. demonstrated that PD-L1 was expressed at significantly higher levels in both the ectopic and eutopic endometrium of patients with EMS compared to healthy endometrial tissues. Moreover, using flow cytometry, they showed that the PD-L1 expression was upregulated in CD8^+^ and CD4^+^ T cells in the PB of EMS patients. The authors observed that 17β-estradiol (E2) upregulates the PD-L1 expression in endometrial epithelial cells [44]. Similarly, Walankiewicz et al. described an increased number of CD8^+^ and CD4^+^ T cells and CD19^+^ B cells with PD-L1 expression in the PB of EMS patients in comparison to the healthy subjects [45].

Our study showed the accumulation of sPD-L1 and sPD-L2 in the PF of EMS patients. The concentration of sPD-L1 in the plasma of EMS patients was significantly lower than that detected in the control group. Interestingly, the plasma sPD-L1 concentration detected in patients with late stages of EMS was significantly higher than in the early stages of the disease.

Similarly to our results, Santoso et al. detected a higher concentration of sPD-L1 in PF, compared to the serum, in the cohort of 44 patients suffering from EMS [46]. The authors observed a higher concentration of serum sPD-L1 in the late stage of EMS. Moreover, they showed almost twice higher sPD-L1 levels in the PF of women with EMS-related infertility in comparison to the control group [46]. In contrary to our study, Okşaşoğlu et al. demonstrated significantly higher levels of sPD-L1 in the serum of EMS patients than in the control group [47]. 

Differences in the distribution of sPD-L1 may be the result of different materials studied in our research and the above-cited papers and/or different clinical EMS stages, and cases included in the control group. In our research, we determined the concentration of sPD-L1 in the plasma. Moreover, our control group consisted of healthy women without any gynecological/other disturbances in contrast to Santoso’s control group with a single benign gynecologic disorder related to the fallopian tubes, ovaries, or fibroids [46]. 

The assessment of the soluble form of PD-L2 in the plasma and PF samples obtained from EMS patients was performed for the first time in our study. We observed a greater accumulation of sPD-L2 in PF than in the plasma of EMS patients. Interestingly, the level of sPD-L2 in PF was 1.22 higher than in the plasma. The concentration of sPD-L2 in the plasma of EMS patients was 1.09 higher than in the control group. Moreover, the plasma sPD-L2 level was significantly elevated in the patients in the early stages of EMS compared to the control group.

PD-L2 can be overexpressed in solid malignancies and may promote immunosuppression. The interaction between the PD-1 receptor and PD-L2 has given similar results to the connection with PD-L1 as the inhibition of T cell proliferation or cytokine production, and also the cytolysis of T cells. In addition, it is considered an independent marker of poor outcomes during esophageal [48], colorectal [49], or pancreatic [50] cancer. Qiao et al. demonstrated that PD-L2 plays a significant role in escaping from immune surveillance in patients with head and neck squamous cell carcinoma (HNSCC) by binding to PD-1-positive tumor-infiltrating lymphocytes (TILs). Moreover, PD-L2 was found to have a positive correlation with lymph node metastasis in HNSCC [51].

The peritoneal cavity is a dynamic microenvironment during the development/progression of EMS, with chronic inflammation and disturbances in the activity of immune cells, such as abnormal lymphocyte response or impaired NK cell-mediated cytotoxicity. The peritoneal cavity environment develops tolerance to endometrial implants by the presence of tolerogenic Tregs [32] or immunosuppressive myeloid-derived suppressor cells (MDSCs) [7]. This is similar to the peritoneal environment during OC development. On the other hand, the activity of CD8^+^ and CD4^+^ T cells is decreased in the PF of EMS patients. Reduced T-lymphocyte response to autologous endometrial cells was also observed in EMS. Disturbances in the immune response may be the key orchestrator of the proliferation of endometrial implants and may lead to avoiding apoptosis and promoting their immunosurveillance from the immune system [13]. In healthy conditions, the PD-1/PD-L1 pathway is responsible for the maintenance of peripheral T cell tolerance in tissues and for protecting them from autoimmunity by inhibiting the activation of self-reactive T cells [52]. Prolonged exposure to endometrial antigens, which occurs in the peritoneal cavity of EMS patients, may lead to an increased expression of PD-1 on T cells and may cause T-cell exhaustion [53]. It is known that the PD-1/PD-L1 pathway may also be responsible for modifying the metabolic activity of effector T cells by inhibiting other pathways, such as Ras/MEK/ERK and PI3K/Akt/mTOR. Exhausted T cells exhibit defective proliferative and cytotoxic activity as well as impaired cytokine production, such as interferon gamma (IFN-γ), tumor necrosis factor (TNF), or interleukin 2 (IL-2) [54]. 

We conclude that an elevated percentage of mDCs, an elevated percentage of PD-L1- or PD-L2-positive mDCs and pDCs, and a higher concentration of the soluble form of PD-L1 and PD-L2, as detected in our study in the PF of EMS patients, play an important role in the modulation of immune response in EMS (Figure 11). A higher concentration of PD-L2, which was detected in PF for the first time in our study, may play a crucial role in inducing immunotolerance and immunosuppression in the microenvironment of the peritoneal cavity in EMS patients, favoring the growth and invasion of endometrial implants.

## 4. Materials and Methods

### 4.1. Patients and the Control Group

The study group consisted of 72 patients with EMS, diagnosed and operated in the 1st Department of Oncological Gynaecology and Gynaecology Independent Public Clinical Hospital No. 1, Lublin, Poland. The control group was obtained thanks to the Regional Centre of Blood Donation and Blood Treatment in Lublin and it consisted of 20 healthy subjects without gynecological disorders. Clinical characteristics of the EMS patient group are presented in the Table 2.

The criteria adopted for exclusion from the examination included infections, allergy, autoimmune disease, pregnancy, and cancer. The conducted research received the approval of the Bioethics Committee at the Medical University of Lublin (KE-0254/171/2021, KE-0254/116/2019). All patients gave their written and oral informed consent to using their material for scientific purposes and signed the written acquiescence form to participation in the study.

### 4.2. Material

The examined clinical material was obtained by the drawing of PB and PF from patients suffering from EMS, and the drawing of PB from the control group. The blood was collected into 9 mL potassium EDTA S-Monovette (SARSTEDT, Germany) before performing laparoscopy.

More specifically, PF was collected aseptically into 15 mL sterile polypropylene Falcon centrifuge tubes (Wuxi NEST Biotechnology, Wuxi, China) during laparoscopy. The mononuclear cells from PB (PB MNCs) and PF (PF MNCs) were isolated by means of density gradient centrifugation. The blood was diluted at a ratio of 1:1 in 0.9% buffered physiological saline-PBS without calcium (Ca^2+^) and magnesium (Mg^2+^) (PAA Laboratories GmbH, Pasching, Austria), and transferred into 3 mL of Gradisol L with a specific gravity of 1.077 g/m (Aqua Medica, Białystok, Poland). PF was transferred without dilution into 3 mL Gradisol L. The gradient centrifugation was performed at an acceleration of 700× *g* for 20 min at room temperature. The PB MNCs and PF MNCs fractions were collected using the Pasteur pipette from the interphase between two layers and washed twice in the PBS buffer without Ca^2+^ and Mg^2+^ for 5 min. The sediment after the last washing was suspended in 1 mL of the PBS buffer without Ca^2+^ and Mg^2+^. Then, the vitality of cells was examined by using 470 µL of PBS, a 25 µL cells suspension, and a 5 µL Trypan Blue solution (Sigma-Aldrich, St. Louis, MI, USA), transferred and counted in a Neubauer chamber. Mononuclear cells were isolated within 2 h of the draw and used for flow cytometry analysis.

### 4.3. Flow Cytometry Analysis

Following their isolation, the mononuclear cells in an amount of 1 × 10^6^ were distributed into tubes and incubated for 20 min at room temperature with a combination of the following monoclonal antibodies (mAbs): anti-BDCA-1 FITC (anti-CD1c; clone: AD5-8E7, cat: 130-113-301; MACS MilenyiBiotec, Bergisch Gladbach, Germany), anti-BDCA-2 FITC (anti-CD303; clone: AC144, cat:130-113-192; MACS MilenyiBiotec), anti-CD19 PE-Cy5 (clone: SJ25C1, cat: 566396; BD Bioscience, Franklin Lakes, NJ, USA), anti-PD-L1 PE-Cy7 (clone: MIH18, cat: 329718; BioLegend, San Diego, CA, USA), anti-PD-L2 PE (clone: 24F.10C12, cat: 329606; BioLegend, San Diego, CA, USA), anti-CD123 PE-Cy7 (clone: 6H6, cat: 306010; BioLegend, San Diego, CA, USA), and anti-PD-L1 APC (clone: MIH1, cat: 563741; BioLegend, San Diego, CA, USA). In the next step, the cells were washed twice with PBS and analyzed as myeloid (BDCA-1^+^CD19^−^) and plasmacytoid (BDCA-2^+^CD123^+^) DCs with PD-L1 or PD-L2 expression through flow cytometry (FACSCanto I Becton Dickinson, Franklin Lakes, NJ, USA) using FacsDiva software. The frequency of mDCs and pDCs are presented as the percentage of mononuclear cells (MNCs). The 100,000 events were acquired for each analysis. The expression levels of PD-L1/PD-L2 are presented as the percentage of total respective cell subsets (i.e., myeloid BDCA-1^+^CD19^−^, plasmacytoid BDCA-2^+^CD123^+^ DCs), which we described in our earlier published manuscript [21]. Fluorescence minus one (FMO) control was used to verify the staining specificity and as a guide for setting the markers to delineate positive populations. The method of identification of pDCs with PD-L1/PD-L2 expression is presented in Figure 12. 

(a)Next, the P1 gated PF MNCs were analyzed for BDCA-1^+^CD19^−^ (region Q4; (B)). The final dot plots were drawn of PD-L1 mDCs (region Q2-1, (C2)) or PD-L2 mDCs (region Q2-2, (D2)). Fluorescence minus one (FMO) control ((C1, D1), respectively) was used to verify the staining specificity and as a guide for setting the markers to delineate positive populations.(b)The P1 gated PB MNCs were analyzed for BDCA-2^+^CD123^+^ (region Q4; (B)). Final dot plots were drawn of PD-L1 pDCs (region Q2-1, (C2)) and PD-L2 pDCs (region Q2-2, (D2)). Fluorescence minus one (FMO) control ((C1, D1), respectively) was used to verify the staining specificity and as a guide for setting the markers to delineate positive populations.

### 4.4. ELISA

The concentration of the soluble forms of PD-L1 (cat: DB7H10; R&D Systems, Minneapolis, MN, USA; detection range: 25 to 1600 pg/mL; sensitivity: 4.52 pg/mL) and PD-L2 (cat: BMS2215; Invitrogen, Waltham, MA, USA; detection range: 39 to 2500 pg/mL; sensitivity: 3.65 pg/mL) in the plasma and PF were detected using the immunoassay kits following the manufacturer’s protocol. All samples were assayed in duplicate. The absorbance of the plate was read on an ELX-800 plate reader (BioTek Instruments, Inc., Winooski, VT, USA) and analyzed using Gen5 (BioTek Instruments, Inc.).

### 4.5. Statistical Analysis

The obtained results were analyzed by means of Statistica 12.0 PL. The Wilcoxon paired test was chosen to compare the results from PB and PF. The Mann–Whitney U test was used with respect to the results of the statistical comparison between the studied groups (the EMS group/the control group). Relationships between the two parameters were investigated using Spearman’s rank correlation test. The data were presented as median, minimum, and maximum values. A *p*-value of <0.05 was considered statistically significant.

## 5. Conclusions

In summary, we demonstrated that the percentages of mDCs, the percentages of both mDCs and pDCs with PD-L1 or PD-L2 expression, and the soluble forms of PD-L1 and PD-L2 are upregulated in the local peritoneal environment of patients suffering from EMS. Even though EMS is a systemic disease with disturbances in the quantity and activity of immune cells, we observed that the peritoneal cavity microenvironment may play an overarching role in this disease. DCs with the expression of PD-L1 and PD-L2 may be connected with immune dysfunction mediated by the PD-1/PD-L1/PD-L2 pathways. It may lead to hampering the activity of T cells and impairing the removal of endometrial implants.

In light of the above-mentioned data, we can conclude that the peritoneal cavity environment and the PD-1/PD-L1/PD-L2 pathways may play an important role in the development and/or progression of EMS.

## Figures and Tables

**Figure 1 ijms-23-11607-f001:**
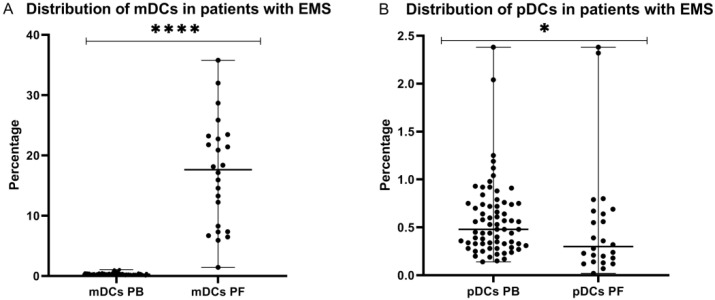
Percentage of mDCs (**A**) and pDCs (**B**) in peripheral blood and peritoneal fluid in patients with endometriosis. The median values with the following signs indicate statistically significant differences: * *p* < 0.05, **** *p* < 0.0001.

**Figure 2 ijms-23-11607-f002:**
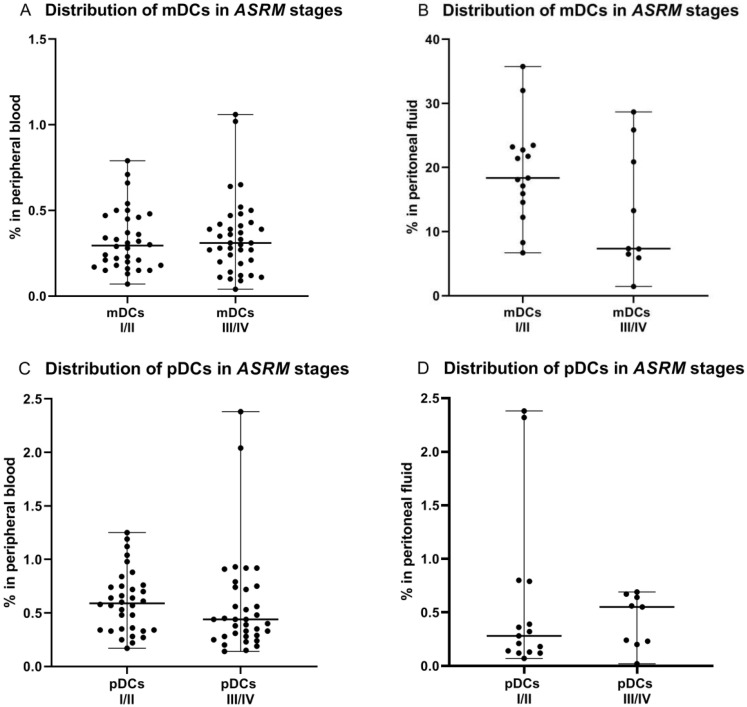
Percentage of mDCs and pDCs in peripheral blood and peritoneal fluid in early (I/II) and late (III/IV) *ASRM* stages of endometriosis (**A**–**D**).

**Figure 3 ijms-23-11607-f003:**
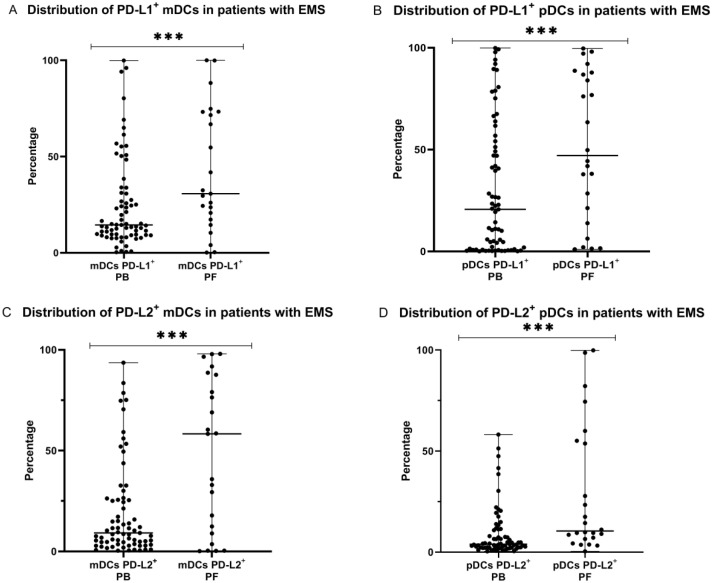
Percentage of mDCs and pDCs with PD-L1 and PD-L2 expression in peripheral blood and peritoneal fluid (**A**–**D**) in patients with endometriosis. The median values with the following sign indicate statistically significant difference: *** *p* < 0.001.

**Figure 4 ijms-23-11607-f004:**
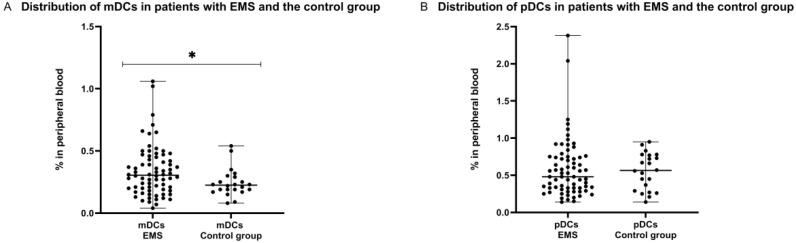
Percentage of mDCs (**A**) and pDCs cells (**B**) in peripheral blood in patients with endometriosis and control group. The median values with the following sign indicate statistically significant differences: * *p* < 0.05.

**Figure 5 ijms-23-11607-f005:**
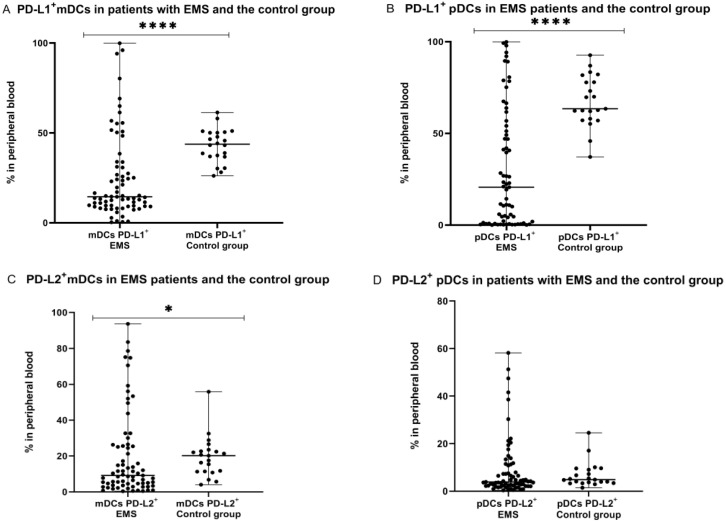
Percentage of mDCs and pDCs with PD-L1 and PD-L2 expression in peripheral blood in endometriosis patients and the control group (**A**–**D**). The median values with the following signs indicate statistically significant differences: * *p* < 0.05, **** *p* < 0.0001.

**Figure 6 ijms-23-11607-f006:**
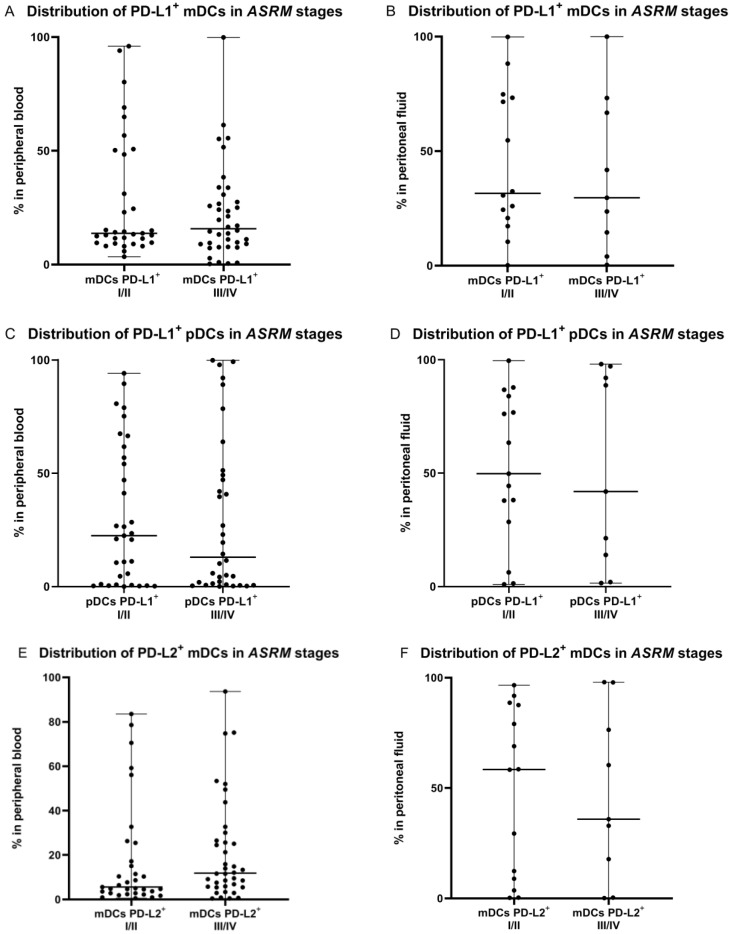
Percentage of mDCs and pDCs with PD-L1 and PD-L2 expression in peripheral blood and peritoneal fluid in early (I/II) and late (III/IV) *ASRM* stages of endometriosis (**A**–**H**). The median values with the following sign indicate statistically significant differences: * *p* < 0.05.

**Figure 7 ijms-23-11607-f007:**
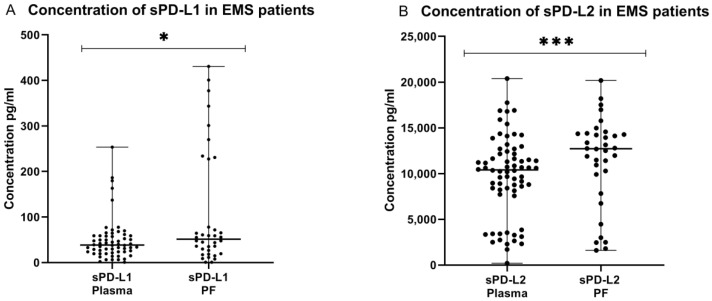
Levels of sPD-L1 (**A**) pg/mL and sPD-L2 (**B**) pg/mL in the plasma and peritoneal fluid of patients with endometriosis. The median values with the following signs indicate statistically significant differences: * *p* < 0.05, *** *p* < 0.001.

**Figure 8 ijms-23-11607-f008:**
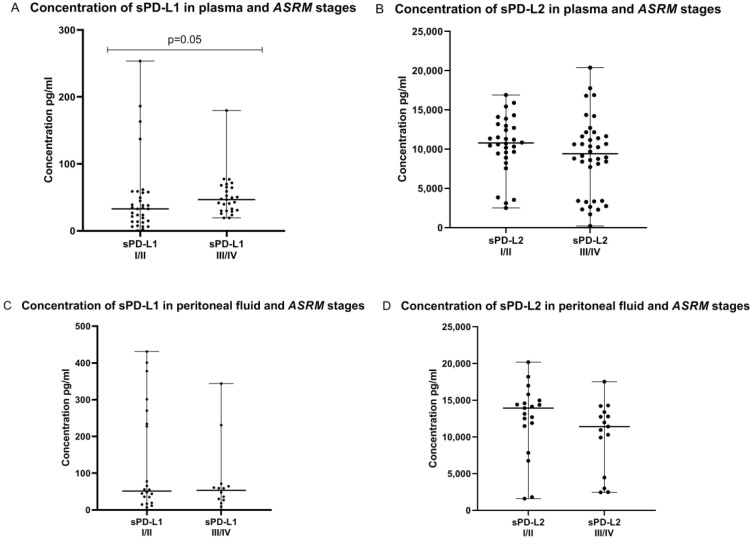
Levels of sPD-L1 and sPD-L2 in the plasma and peritoneal fluid in early (I/II) and late (III/IV) *ASRM* stages of endometriosis (**A**–**D**) pg/mL.

**Figure 9 ijms-23-11607-f009:**
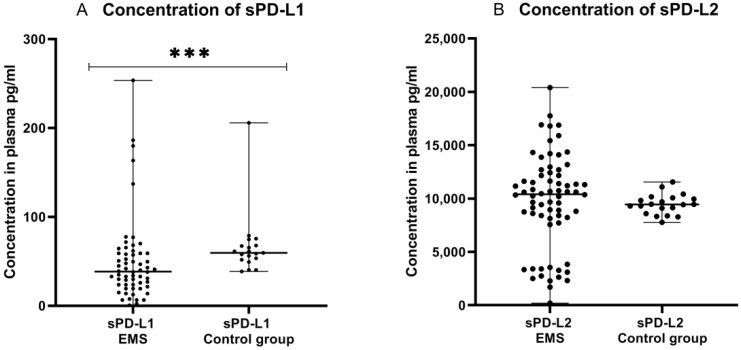
Levels of sPD-L1 (**A**) pg/mL and sPD-L2 (**B**) pg/mL in the plasma of patients with endometriosis and control group. The median values with the following sign indicate statistically significant differences: *** *p* < 0.001.

**Figure 10 ijms-23-11607-f010:**
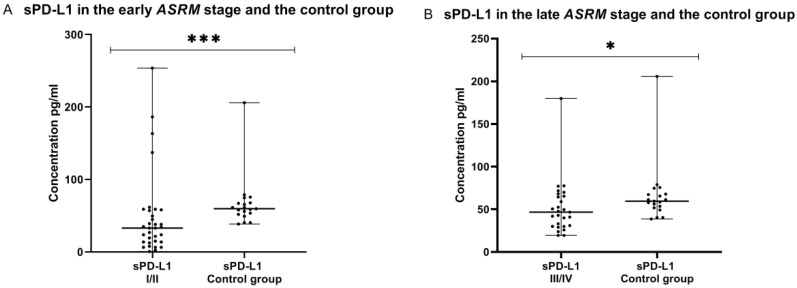
Levels of sPD-L1 and sPD-L2 in the plasma in early (I/II) and late (III/IV) *ASRM* stages of endometriosis and in the plasma of the control group (**A**–**D**) pg/mL. The median values with the following signs indicate statistically significant differences: * *p* < 0.05, *** *p* < 0.001.

**Figure 11 ijms-23-11607-f011:**
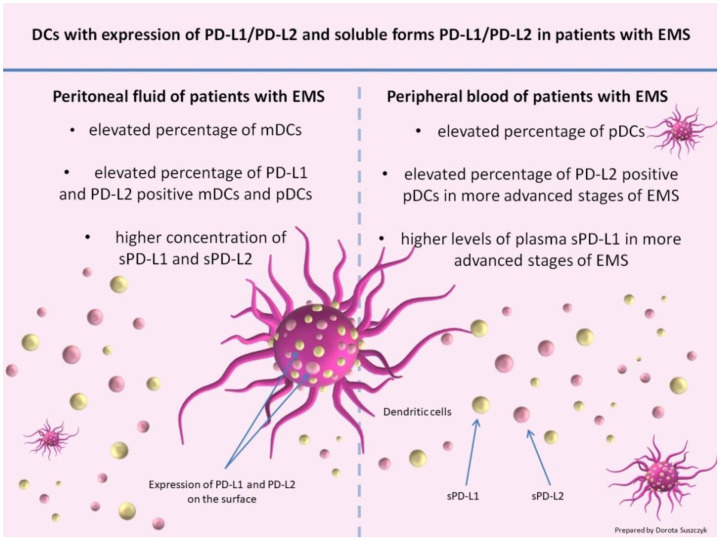
Summary of the most important points of the study.

**Figure 12 ijms-23-11607-f012:**
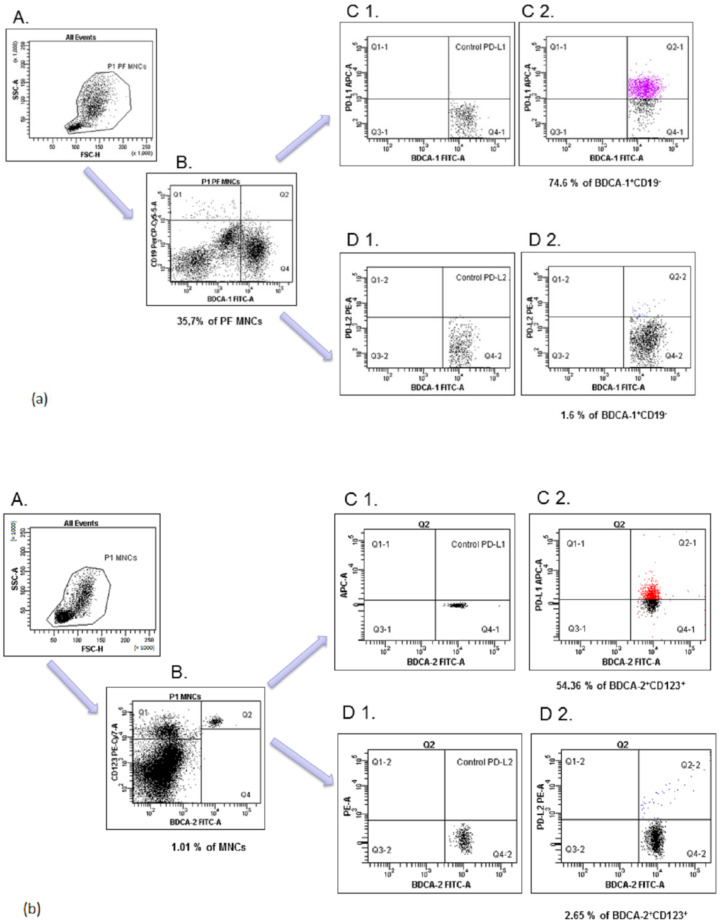
Flow cytometric analysis of BDCA-1^+^CD19^−^ and BDCA-2^+^CD123^+^ dendritic cells with PD-L1 and PD-L2 expression in the PF (**a**) and PB (**b**) of the patient with EMS. An acquisition gate was established based on FSC and SSC that included mononuclear cells (P1 population; (A)).

**Table 1 ijms-23-11607-t001:** Percentage of PD-L1- and PD-L2-positive dendritic cells in patients with endometriosis and the control group.

	Peripheral Blood EMS		Peritoneal Fluid EMS		Peripheral Blood Control Group	
	Median	Range	Median	Range	Median	Range
%BDCA-1^+^CD19^−^ (mDCs)	0.31	0.04–1.06	17.63	1.44–35.78	0.22	0.08–0.54
%BDCA-2^+^CD123^+^ (pDCs)	0.48	0.14–2.38	0.30	0.02–2.38	0.56	0.14–0.95
%BDCA-1^+^CD19^−^PD-L1^+^	14.46	0.29–99.87	30.69	0.16–99.95	43.69	26.11–61.35
%BDCA-2^+^CD123^+^PD-L1^+^	20.65	0.05–99.87	47.06	1.01–99.63	63.44	37.14–92.65
%BDCA-1^+^CD19^−^PD-L2^+^	9.14	0.39–93.64	58.31	0.16–97.96	20.21	4.03–55.80
%BDCA-2^+^CD123^+^PD-L2^+^	3.89	0.36–58.22	10.47	0.34–99.88	4.86	1.44–24.60

EMS—endometriosis.

**Table 2 ijms-23-11607-t002:** Clinical characteristics of the EMS patient group.

Clinical Feature	EMS Patients (*n* = 72)
Age (median), years (range)	31.5; 19–48
Stages of EMS (the *ASRM* classification system)	
Early (I/II) *n* = 34	
Stage I (minimal)	15
Stage II (mild)	19
Advanced (III/IV) *n* = 38	
Stage III (moderate)	29
Stage IV (severe)	9
	Healthy subjects (*n* = 20)
Age (median), years (range)	28; (20–38)

## Data Availability

All data generated or analyzed during this study are included in this publication.

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
