# Peer review of "Clinical Value of the PD-1/PD-L1/PD-L2 Pathway in Patients Suffering from Endometriosis"

_ijms, 2022, doi:10.3390/ijms231911607_

Round 1

Reviewer 1 Report

This manuscript is very interesting, generally well written and well illustrated. Only few points deserve to be improved. In particular: 

Line 47: It is important to point out that endometriotic lesions are also characterized by a high proliferative rate that is also stimulated by the inflammatory microenvironment (PMID: 26708185)

 Product codes must be reported

Author Response

Reviewer 1

Dear Reviewer,

            We would like to kindly thank you for your prompt and valuable review of our paper. We have addressed the comments raised by you and appropriate revision has been made in the manuscript. Point-by-point responses to the reviewer’s comments are listed below this letter. We are truly hoping that now the paper will fulfil your requirements.

“This manuscript is very interesting, generally well written and well illustrated. Only few points deserve to be improved. In particular: 

Line 47: It is important to point out that endometriotic lesions are also characterized by a high proliferative rate that is also stimulated by the inflammatory microenvironment (PMID: 26708185)”

Response: Thank you very much for this important comment. We have included this information and the recommended reference in the revised manuscript. We agree that it will help to ensure a higher quality of our manuscript and make it more interesting.

“Product codes must be reported”

Response: Thank you very much for this suggestion. According to Reviewer 1 suggestion, we have included codes of products in the revised manuscript.

Reviewer 2 Report

I have read the article “Clinical value of the PD-1/PD-L1/PD-L2 pathway in patients suffering from endometriosis” with great interest. The article describes that there is an increase of mDCs and pDCs in the peritoneal fluid of women with endometriosis when compared to peripheral blood. It also describes the concentration of soluble forms of PD-L1 and PD-L2 in both PF and PB, which could have implications in the immune dysfunction of Treg, favoring the implantation of endometrial tissue in ectopic sites and development of the endometriotic lesions. I have some comments.

Major comments:

1. The design of the study does not include samples in different phases of the menstrual cycle, or the authors have not described this. As endometriosis is an estrogen-dependent disease, this is very relevant as the results may vary extremely between phases of the menstrual cycle.  

Minor comments:

2. In the introduction, the authors put too much emphasis in ovarian cancer while this is not the focus of the article. Maybe they should reduce it a little.

3. Regarding the structure of the results, it would be more understandable for the reader if the results of EMS vs Ctrl were exposed before than the comparisons between stages of EMS. 

4. In Figure 1, the percentage of mDCs in PF seems very high (almost 40%). Is this from the total immune cells? Can the authors specify which are the total cells? If so, do the authors think that it is a reliable result to obtain such high percentage of mDCs? 

5. In Figure 3 and 4, the Y axis show a percentage that goes all the way to 150. Maybe they could lower the percentage to 100 or 110, so the lower percentages could be better appreciated. 

6. The authors stated that pDCs are lower in EMS patients than in the control group in PB. However, there is literature that explains the contrary (BMC Med. 2022 Apr 15;20(1):158. doi: 10.1186/s12916-022-02359-4.). They should discuss these contrary results. It might be due to the menstrual cycle phase, different patients used… 

7. In Table 1, the authors show the markers that they used to isolate the cells. However, the marker strategy that has been used is not explained anywhere in the paper and they just show the markers in the table. They should explain the gating strategy in MM. In addition, in Table 1 they should indicate which cells are the ones expressing those markers: e.g., BDCA-2+CD123+ (pDCs), as not everyone is familiarized with the markers. 

8. Why didn’t the authors measure the sPD-L2 between stages of EMS? 

9. Figure 9: It seems that there are outliers. It is very difficult to see the difference in concentration of sPD-L1 between EMS and Ctrl. I would recommend deleting the outliers and perform the statistics again. At least, they could cut the Y axis, so the bottom dots can be seen.

10. In the graph in Figure 9 it seems that the Ctrl group has higher concentration while in the text the authors state that the concentration is higher in EMS than in controls. Is this a mistake in the text? 

11. In line 305, the authors state that EMS is considered an autoimmune disease, while this is not totally true. Some authors consider it, but it has not been proven. 

12. In MM, the authors do not mention that the control group patients have not any gynecological disorder. I could see it later in the discussion, but they should also mention it in MM. 

13. Figure 11 is not mentioned in the text. 

14. I see that the age of EMS patients goes up to 48 years old. Have the authors confirmed that these patients are not menopausal? If so, they should also mention it in MM. 

15. Resolution of Figure 12 is not very good.

16. In Figure 12 C, it seems that there are two populations in the left-bottom gate. They took all the cells in this gate and then gated for BDCA-2 and PDL-1. It looks to me that they are also including CD123- cells in this gate, and thus these cells are not pDCs. Could the authors explain the gating strategy better? Probably in MM.

Author Response

Dear Reviewer,

            We would like to kindly thank you for your prompt and valuable review of our paper. We have addressed the comments raised by you and appropriate revision has been made to the manuscript. Point-by-point responses to the reviewer’s comments are listed below this letter. We are truly hoping that now the paper will fulfil your requirements.

”I have read the article “Clinical value of the PD-1/PD-L1/PD-L2 pathway in patients suffering from endometriosis” with great interest. The article describes that there is an increase of mDCs and pDCs in the peritoneal fluid of women with endometriosis when compared to peripheral blood. It also describes the concentration of soluble forms of PD-L1 and PD-L2 in both PF and PB, which could have implications in the immune dysfunction of Treg, favoring the implantation of endometrial tissue in ectopic sites and development of the endometriotic lesions. I have some comments.

Major comments:

  1. The design of the study does not include samples in different phases of the menstrual cycle, or the authors have not described this. As endometriosis is an estrogen-dependent disease, this is very relevant as the results may vary extremely between phases of the menstrual cycle.”

Thank you very much for drawing our attention to the phases of the menstrual cycle. The design of our study aims to compare the expression of PD-L1/PD-L2 on myeloid DCs (mDCs) and plasmacytoid DCs (pDCs) and focus on differences in two environments: peripheral blood (PB) and peritoneal fluid (PF), and ASRM stages of endometriosis. It would be a great idea to compare levels of subpopulations in the same patient, in the same milieu focusing on different phases of the menstrual cycle. Unfortunately, one of the important points is that the peritoneal fluid is collected only once, during laparoscopy and it is impossible to take PF from the same patient during different phases of the menstrual cycle. The presented proposal requires a different design of the study and focuses on the other problem in the pathogenesis of endometriosis like hormonal status/phases/concentration of hormones. We appreciate your suggestion and we are concerned that it is an excellent starting point for future research. We will take this suggestion to prepare the original article in the future. This study will aim to focus on the hormonal status/ phases/concentration of hormones in patients with EMS.

Minor comments:

  1. In the introduction, the authors put too much emphasis in ovarian cancer while this is not the focus of the article. Maybe they should reduce it a little.

Response: Thank you very much for this remark. According to your suggestion, we have reduced information about ovarian cancer in the “Introduction” section.

  1. Regarding the structure of the results, it would be more understandable for the reader if the results of EMS vs Ctrl were exposed before than the comparisons between stages of EMS. 

Response: Thank you very much for this important suggestion. We have changed the order of results of EMS vs. control group before the comparisons between early and late stages of EMS patients. We agree that it will help to ensure a higher quality of our manuscript and make it more understandable.

  1. In Figure 1, the percentage of mDCs in PF seems very high (almost 40%). Is this from the total immune cells? Can the authors specify which are the total cells? If so, do the authors think that it is a reliable result to obtain such high percentage of mDCs? 

Response: Thank you very much for this important suggestion. The frequency of mDCs and pDCs are presented as the percentage of mononuclear cells (MNCs). We agree with this point and we added the description to the section “Material and methods”.

  1. In Figure 3 and 4, the Y axis show a percentage that goes all the way to 150. Maybe they could lower the percentage to 100 or 110, so the lower percentages could be better appreciated. 

Response: Thank you very much for this important point. We have changed the percentage to 100 to prepare more readable figures.

  1. The authors stated that pDCs are lower in EMS patients than in the control group in PB. However, there is literature that explains the contrary (BMC Med. 2022 Apr 15;20(1):158. doi: 10.1186/s12916-022-02359-4.). They should discuss these contrary results. It might be due to the menstrual cycle phase, different patients used… 

Response: Thank you very much for focusing our attention on this publication. According to your suggestion, we discuss the results in the “Discussion” section. “On the contrary, Vallvé-Juanico et al. using mass cytometry, showed a higher proportion of pDCs in the peripheral blood of women suffering from EMS than in the control group. These contrary results might be a due assessment of pDCs in the proliferative phase of the menstrual cycle by the cited authors, and also different monoclonal antibodies used for phenotyping, or different cohorts of the patients. It should be stressed that the cited authors evaluated pDCs in a very small study group (n=13) of EMS patients versus the control group (n=6). In our study, we evaluated DCs in numerous groups of patients with EMS (n=62) and healthy women (n=20). [Vallvé-Juanico;2022]”

  1. In Table 1, the authors show the markers that they used to isolate the cells. However, the marker strategy that has been used is not explained anywhere in the paper and they just show the markers in the table. They should explain the gating strategy in MM. In addition, in Table 1 they should indicate which cells are the ones expressing those markers: e.g., BDCA-2+CD123+ (pDCs), as not everyone is familiarized with the markers. 

Response: Thank you for this valuable remark. We have decided to add information about cell markers (BDCA-1+CD19- for mDCs, BDCA-2+CD123+ for pDCs) to Table 1. to make it more readable for researchers and increase its overall comprehensibility.

  1. Why didn’t the authors measure the sPD-L2 between stages of EMS? 

Response: Thank you for this comment. We measured the concentration of sPD-L2 between the early and late stages of EMS and the results are presented in Figure 8.

  1. Figure 9: It seems that there are outliers. It is very difficult to see the difference in concentration of sPD-L1 between EMS and Ctrl. I would recommend deleting the outliers and perform the statistics again. At least, they could cut the Y axis, so the bottom dots can be seen.

Response: Thank you very much for this important suggestion. We strongly agree with this point. We deleted one outlier from the study group (concentration of sPD-L1 789,1 pg/ml in plasma, and 553,0 pg/ml in PF), one from the control group (concentration of sPD-L1 667,2 pg/ml in plasma), and we performed the statistics again. We received new results which are described below. Moreover, we corrected all Figures with a soluble form of sPD-L1.

  • The concentration of sPD-L1 in the peritoneal fluid (n=36) was significantly higher (p<0.05) than in the plasma (n=61) of EMS patients (Figure 7A.). Moreover, we included information about decreased numbers of samples in brackets.
  • We demonstrated in Figure 8A. that the plasma sPD-L1 concentration in patients with late stages (III/IV) of endometriosis was higher than in early (I/II) stages of the disease and the difference reached the level of significance (p=0.05).
  • The significance level of concentration of sPD-L1 in EMS patients and the control group (Figure 9A.) did not change and it was significantly lower (p<0.001) in the plasma (n=61) of EMS patients than detected in the control group (n=19). We included information about decreased numbers of samples in brackets.
  • The concentration of both plasma sPD-L1 in the early (I/II) and late (III/IV) ASRM stages of EMS was significantly lower (p<0.001 and p<0.05, respectively) than detected in the control group (Figure 10A. and Figure 10B.).

  1. In the graph in Figure 9 it seems that the Ctrl group has higher concentration while in the text the authors state that the concentration is higher in EMS than in controls. Is this a mistake in the text? 

Response: Thank you very much for this important point. It was a mistake in the text. We have performed the statistics again to check the concentration of sPD-L1 in the EMS patients group and control. We confirm that “The concentration of sPD-L1 was significantly lower (p<0.001) in the plasma of EMS patients than detected in the control group.” This is correct now.

  1. In line 305, the authors state that EMS is considered an autoimmune disease, while this is not totally true. Some authors consider it, but it has not been proven.

Response: Thank you for this interesting point in the discussion. In our article, we decided to present the point that “endometriosis can be considered an autoimmune disease” as one of the possible ways in the pathogenesis of the EMS. The main aetiology of endometriosis is still unsolved and many authors discussed disturbances in the immune system with the induction of autoimmunity  (PMID: 12763528, PMID: 22330229).  Moreover, other researchers in the systematic review of the observational population study confirmed the hypotheses about the link between endometriosis and autoimmune diseases like SLE, SS, RA, ATD, CLD, MS, IBD, and Addison’s disease (PMID: 31260048). The evidence mentioned above supports the idea that endometriosis may be considered an autoimmune disorder with an autoimmune background, however, it needs further studies. Therefore we change our statement in the manuscript as follows: “According to some studies, EMS has some features an autoimmune disease manifested by tissue damage and the production of autoantibodies (against endometrium, histones, ovary, and phospholipids), and may be associated with other autoimmune diseases.”

  1. In MM, the authors do not mention that the control group patients have not any gynecological disorder. I could see it later in the discussion, but they should also mention it in MM. 

Response: Thank you for this valuable remark. According to the suggestion, we added information about the absence of gynaecological disorders in the control group to the “Material and methods” section.

  1. Figure 11 is not mentioned in the text. 

Response: According to your suggestion, we mentioned Figure 11. in the “Discussion” section.

  1. I see that the age of EMS patients goes up to 48 years old. Have the authors confirmed that these patients are not menopausal? If so, they should also mention it in MM.

Response: We confirm that patient on the day of the experiment has been regularly menstruating and did not declare symptoms of menopause.

  1. Resolution of Figure 12 is not very good.

Response: Thank you very much for this important suggestion. We improved the resolution of Figure 12. 

  1. In Figure 12 C, it seems that there are two populations in the left-bottom gate. They took all the cells in this gate and then gated them for BDCA-2 and PDL-1. It looks to me that they are also including CD123- cells in this gate, and thus these cells are not pDCs. Could the authors explain the gating strategy better? Probably in MM.

Response: Thank you for this opinion. According to your suggestion, we explain the gating strategy better in MM. We hope it is more readable and clear now. The frequency of mDCs and pDCs are presented as the percentage of mononuclear cells. For each tube, 100,000 events were acquired and analyzed using FacsDiva software. The expression levels of PD-L1/PD-L2 are presented as the percentage of total respective cell subsets (i.e. myeloid BDCA-1+CD19-, plasmacytoid BDCA-2+CD123+ DCs), which we described in our earlier published manuscript (PMID: 34768993) . We added fluorescence minus one (FMO) control which was used to verify the staining specificity and as a guide for setting the markers to delineate positive populations to clarify the gating strategy better. The method of identification of mDCs and pDCs with PD-L1/PD-L2 expression is presented in Figure 12.

Round 2

Reviewer 2 Report

The authors have answered all my questions and have made the suggested edits to the article.